# Relationship of Training Factors and Resilience with Injuries in Ski Mountaineers

**DOI:** 10.3390/sports10120191

**Published:** 2022-11-25

**Authors:** Paula Etayo-Urtasun, Patxi León-Guereño, Iker Sáez, Arkaitz Castañeda-Babarro

**Affiliations:** 1Department of Physical Activity and Sport Science, Faculty of Education and Sport, University of Deusto, 48007 Bilbao, Bizkaia, Spain; 2Department of Physical Activity and Sport Science, Faculty of Education and Sport, University of Deusto, 20012 Donostia-San Sebastian, Gipuzkoa, Spain

**Keywords:** ski mountaineering, skimo, injury, prevention, core training, warm up, resilience, equipment

## Abstract

Ski mountaineering is an increasingly popular sport with a relatively high risk of injury. Therefore, several studies have analyzed factors related to the likelihood of injury, including athlete characteristics, training, resilience and equipment. Thus, the aim of this study was to identify factors that may influence injury risk. A 15-minute online survey was sent to various ski mountaineering groups of different ages and levels. Both the Mann–Whitney U test and odds ratio analysis were performed in order to analyze the data. Results from 104 skiers showed that most injuries occurred in the lower extremities, especially in the knee (29.6%). The findings indicated that skiers who had suffered an injury performed in more competitions per year (*p* = 0.046), more ski mountaineering sessions per week (*p* = 0.022) and fewer core training sessions per week (0.029), although core training and competition were not statistically significant factors. Likewise, non-injured athletes had fewer pairs of skis (*p* = 0.019), which were also wider (*p* = 0.04). However, no difference was found for warm up and resilience between both groups (*p* = 0.275). In conclusion, it is important to implement preventive measures based on these factors, even if more research is needed.

## 1. Introduction

Ski mountaineering is an increasingly popular sport and has been officially included in the upcoming 2026 Winter Olympics in Milano-Cortina, which will contribute to an increase in the number of competitions and higher training volumes [1,2]. This winter sport consists of downhill and uphill sections, and it includes different race formats according to the pattern of the slopes, as well as the length and total vertical gain [2,3,4].

Ski mountaineering demands a high level of physical fitness, and it is characterized by a significant cardiopulmonary strain due to the hypoxic environmental conditions in high altitudes [3,4]. Therefore, maximal oxygen uptake (VO_2_max) and oxygen uptake at the second ventilatory threshold (VT_2_) seem to be key factors in performance [1]. In addition, it is important to achieve the optimal balance between muscle power and body mass in order to achieve efficient muscular performance at high intensities [2,3].

Snow sports involve high speeds as well as great levels of resistance, which means that injuries are often serious and can affect the competitive status and quality of life of the athlete [5]. As an example, during the 2020 Youth Olympic Winter Games (YOG 2020), 12% of athletes sustained at least one injury [6]. Although ski mountaineering was ranked among the winter sports with the lowest incidence of injury [6], the number of injuries is expected to considerably increase due to the further professionalization of the sport [2]. According to the data obtained during YOG 2020, the most common injury was to the knee, followed by the head and the hand [6]. However, acute injuries present a non-negligible risk that may further increase due to the future professionalization of the sport [2].

Therefore, in recent years there has been remarkable interest in factors which may reduce the risk of injury. As in other sports, uncertain environments such as mountains are associated with a large number of threats [1,7]. Therefore, it is very important to follow mountain safety recommendations and to wear enough protective equipment [8]. However, in this article, we will focus on training-related aspects, targeting those factors that are modifiable from the perspective of coaches and trainees.

Regarding the training routine, physical exercise consisting of a structured neuromuscular warm-up for injury prevention could have an immediate effect in improving the activation time of the knee stabilizer muscles, thus potentially reducing the risk of injury to the knee [9]. Furthermore, the meta-analysis by Ding et al. [10] demonstrated that performing an adequate warm-up can help to significantly reduce the risk of injury by as much as 36%.

Another factor related to training would be strength training, which may be effective to prevent acute and overuse injuries in many sports. In fact, the meta-analysis conducted by Lauersen et al. [11] found that strength training programs may reduce sports injuries by an average of 66%. As an example, Raya-González et al. [12] found that a strength training program may be useful in reducing injury incidence in young soccer players. Regarding winter sports, it seems that core strength is particularly relevant. In this line, Raschner et al. [13] found an increased ACL injury risk for those alpine skiers with low core strength. Therefore, it has been suggested that core strength training could be an effective strategy to reduce the likelihood of injury [8,14].

In addition, it seems that equipment may also influence the incidence of ski injuries [14]. In alpine skiing, for instance, it seems that not only the protective equipment, but also the characteristics of the skis can play an important role in injury prevention [8]. Similarly, it appears that ability, skiing level, fitness and experience also play a significant role in the risk of injury [14].

Lastly, there are psychological factors which can significantly affect the risk of injury. Resilience, defined as the ability to positively adapt to stressful situations, seems to have a negative relationship with the number of injuries in amateur runners. In other words, the most resilient athletes suffered fewer injuries than those with lower resilience [15].

Nevertheless, it remains unclear as to which factors are relevant to the risk of injury in ski mountaineering, as most studies have focused on athletes from other sports. Therefore, the main aim of this study is to identify potential modifiable factors that may influence the incidence of injury in ski mountaineering. For this purpose, the participants of this research completed an online survey which was followed by several statistical analyses. As a result, this study will contribute to the development of injury prevention strategies in ski mountaineering.

## 2. Materials and Methods

### 2.1. Sample Characteristics

The investigation was conducted as a cross-sectional survey-based study. Participants were selected by convenience sampling. The only inclusion criterion was that participants had to practice ski mountaineering. Therefore, a survey was designed and sent to several European ski mountaineering groups by WhatsApp or mail. Moreover, the survey was published on social networks. In this way, each participant filled out the questionnaire in a different place and time. There were no exclusion criteria related to age, level, years of experience, body composition or injury. The only exclusion criterion was not completing the questionnaire completely.

Participants received all the necessary information about the study before completing the survey, such as the objective, procedure and confidentiality of the research. Their rights were preserved by requesting their voluntary participation and offering them the opportunity to withdraw from the study at any time. Therefore, all subjects gave their informed consent for inclusion before they participated in the study.

### 2.2. Data Collection Methods

An online survey was used to collect all the necessary information from each subject. The survey consisted of four sections: (I) informed consent and demographics, (II) training and competition, (III) reduced scale of resilience and (IV) injuries (Appendix A). Most of the sections included multiple choice questions. However, section III involved a reduced scale of resilience, a valid and reliable instrument formed by 10 items rated from 0 to 4 [16,17].

The authors designed the first draft of the questionnaire, and a skier conducted a pilot test to detect possible errors or inconsistencies. When the official survey was approved, we sent a letter to several ski mountaineering teams to explain the objective of the study and to ask for their participation. Furthermore, the questionnaire was sent to specific skiers, and it was published on social networks. The survey was administered through a software called Google Forms and took approximately 15–20 min to complete.

### 2.3. Statistical Analysis

In order to test the normality of the variables, the Kolmogorov–Smirnov test (*n* > 50) was performed, which indicated that the data did not follow a normal distribution. Therefore, nonparametric tests were used to analyze the factors influencing the probability of injury. The Mann–Whitney U test for independent samples was performed to analyze the continuous variables according to whether the skiers had suffered an injury or not. These variables were presented as mean and standard deviation. The significance level for this type of statistical analyses was established at 0.05 (*p* ≤ 0.05). In addition, odds ratio analysis was used to examine dichotomous variables. In this case, frequency analyses were performed, and the statistical significance was considered when the range between the lower and upper values did not contain the value of 1. All statistical tests were conducted through IBM SPSS Statistics 28.

## 3. Results

A total of 104 skiers completed the survey, representing a response rate of 100%. Among the participants, 79.8% of them were male and 20.2% were female. Most of them were from Spain. The mean age was 37.21 ± 12.69 years. The 104 responses collected were divided into two groups according to whether the participants had suffered an injury while ski mountaineering. As such, there were 50 skiers who confirmed having suffered an injury, whereas the remaining 54 skiers had not experienced any injury.

### 3.1. Demographic and Physical Characteristics Data

Data related to physical characteristics were not significantly linked to the likelihood of injury during ski mountaineering. However, the number of competitions per year was positively related to the probability of injury (*p* = 0.046), which means that skiers who participate in more competitions seem to have a higher probability of injury (Table 1).

### 3.2. Characteristics of Recorded Injuries

Regarding the characteristics of the injuries analyzed, most of the injuries occurred in the lower extremities, such as knees (29.6%), foot (14%), ankle (10.9%) and leg (9.3%). In addition, most injuries occurred on one of the two sides of the body, with only 4.7% of injuries located near the longitudinal axis.

Concerning the timing of the injuries, the majority of them occurred during training (75%), although some injuries also took place in competitions (18.7%). Moreover, the highest incidence of injuries occurred when the individual was travelling downhill (65.6%).

The most common reason for injuries is usually impact (68.8%), although there are also injuries caused by overuse (32.2%). However, the injuries analyzed had different diagnoses. Therefore, they also had varying recovery periods, ranging from 1–3 days to more than 3 months without skiing (Table 2).

### 3.3. Training Characteristics

In addition, skiers who had suffered an injury had a significantly higher number of ski mountaineering sessions per week (*p* = 0.022), thus suggesting that more ski mountaineering sessions may lead to a higher risk of injury. In addition, the non-injured participants had more core training sessions per week (*p* = 0.029), which emphasizes the importance of core training (Table 3).

The dichotomous variables related to training were examined by odds ratio analysis. No statistical significance was found for any of the variables analyzed. The highest effect corresponds to the fact of competing or not (prevalence of injury among competitors 60% versus 40% of recreational skiers). Therefore, findings suggest that these variables are not decisive factors in terms of injury incidence (Table 4).

### 3.4. Resilience

In terms of psychology, resilience was not significantly different according to whether skiers had suffered an injury or not (*p* = 0.275). Among the items analyzed, significant differences were found for only one of them. Thus, skiers who had suffered an injury reported a significantly higher score when asked about their strength to face challenges and difficulties in life (Table 5).

### 3.5. Equipment

Regarding equipment, injured skiers had significantly more skis than those who had not suffered any injury (*p* = 0.019). In addition, the skis owned by participants who had suffered an injury were significantly narrower than those of non-injured skiers (*p* = 0.040). However, the number of pairs of poles (*p* = 0.068) and their length (*p* = 0.326) were not significantly different between both groups (Table 6).

## 4. Discussion

The main objective of this study was to identify the factors that influence the risk of injury in ski mountaineering. According to the results, skiers who had suffered an injury performed more core training sessions per week and participated in more competitions per year. However, performing core training or participating in competitions did not seem to be decisive factors in the risk of injury. The factors that did appear to be significant were the number of ski sessions per week, the number of pairs of skis and their width.

Regarding the characteristics of the injuries, the most injured part was the lower extremity, particularly the knees (29.6%), which is in line with previous research [5,6,18,19,20]. In addition, most of the injuries (65.6%) occurred during downhill slopes, which could be due to a lack of control caused by the variety of surface conditions [1,2]. It is important to note that although several injuries caused no significant time loss, in some cases (15.6%) the recovery period took more than 3 months.

Even though in other sports strength training seems to be effective in reducing the incidence of injury [11,12], the present study failed to find a relationship between strength training and injury reduction. A possible explanation could be that strength training may help reduce overuse injuries but not impact injuries, which were more frequent in the present sample [11,21]. Another reason might be the quality of training. Therefore, it is not enough to simply undergo training; it is also necessary to strengthen the failure thresholds of the relevant tissues with proper techniques and progression [11].

Based on the results obtained in the present study, the non-injured skiers performed significantly more core training sessions per week, even though core training is not a statistically significant factor. This is not in line with results obtained in other winter sports, for instance, alpine skiing [8,13,22]. In alpine skiing, a lower core strength might increase the tendency towards valgus collapse, leading to an unstable knee position while skiing [13]. In contrast, a well-trained abdominal, spine, back extensor and quadratus lumborum musculature helps to maintain a central position, contributing to cope with external forces during skiing [22,23]. In any case, it is important to note that the technique used in both winter sports is slightly different, especially during downhill skiing, which could explain the different results obtained. Furthermore, it is not possible to rule out the importance of core training in reducing the risk of injury, as core strength could counteract the reduction in control by skiers during descents [1].

According to the findings, warm up was not a statistically significant factor in terms of injury risk. These results are in disagreement with several studies that have shown that warm up intervention programs can lead to a significant reduction in injury rates in several sports [9,10]. Therefore, for a warmup to be effective, it is not enough to simply carry it out; it has to fulfill several requirements, such as not involving weightlifting, as fatigue may increase the risk of injury [11].

Furthermore, ski mountaineers who had been injured participated in more competitions per year, even if taking part in competitions was not a significant factor itself. This is in line with previous studies and seems to make sense considering that competitions are a stressful time in which athletes tend to take more risks [6,8,13,24]. In addition, the number of skiing sessions per week was significantly higher in the group of injured skiers. That seems reasonable considering that ski mountaineering is an endurance sport that requires high energy and large volumes of repetitive training, which could lead to a meaningful increase in overuse injuries [3,4,6,24,25]. Furthermore, as in other snow sports, ski mountaineering faces external dangers such as avalanches, changing snow surfaces and weather changes [1,2,6,26].

Regarding equipment, the skis of the injured skiers were significantly narrower than those of the non-injured skiers (*p* = 0.04). This is not in line with the results of Spörri et al. [8], who reported that wide skis are more likely to cause the knee joint to move unfavorably in the transverse and frontal planes, significantly increasing the risk of injury [27]. In addition, injured skiers had significantly more skis than those who had not suffered any injury (*p* = 0.019). This factor has not been previously studied, but it may be related to the lack of familiarization with the equipment. However, regarding the rest of the equipment-related factors, the number of pairs of poles and their length were not significantly different between both groups.

Regarding resilience, this variable was not significantly different depending on skiers having suffered an injury or not. This disagrees with the results of León-Guereño et al. [15] but supports the findings of Castro Sanchez et al. [28], who found differences that were not significant. Therefore, it could be suggested that even though in some cases injuries are a learning process to better manage stressful situations, in other cases this period of adversity is not adequately faced and as a result, it causes negative consequences [15,28].

The present study has some limitations. The main limitation would be that the data was collected through a retrospective questionnaire and not through a daily monitoring system. Consequently, there are several factors that could be important and have not been analyzed, for instance, physical condition, quantification of training load, season period, ski technique, physical environment conditions or even skiing speed. Moreover, as in retrospective survey studies, individual recall of risk variables could be inaccurate [29]. Furthermore, the sample of the study was small and thus, caution is warranted when generalizing the results. Moreover, when analyzing odds ratios with a larger sample size, the effects of the same size would probably be statistically significant. In addition, research on injury risk in ski mountaineering is scarce, which has made it difficult to compare the findings with those of other studies. It is surprising that so few studies have been carried out on this sport compared to other winter sports such as alpine skiing.

Consequently, further research is needed. On the one hand, it would be advisable to have a larger study cohort in order to generalize the current findings. Furthermore, sex differences in ski mountaineering have been little studied and thus, it may be convenient to study the influence of risk factors according to the sex of the participants. On the other hand, it is important to emphasize the complexity and context-dependence of injury patterns and risk factors in ski mountaineering. Therefore, the complexity of the skiing-specific situation cannot be solved by a single approach and interdisciplinary studies need to be carried out. In addition, it would also be recommendable for future studies to examine the effect of prescribing specific exercises aimed at reducing the risk of injury in ski mountaineering.

## 5. Conclusions

In summary, even if non-injured skiers performed more core training sessions per week and took part in more competitions per year, performing core training and competitions were not significant factors in injury risk, which means that more research is needed. Conducting more ski sessions may lead to an increase in the number of injuries, which is why training schedules need to be carefully designed. Finally, it is important to become used to skis with the correct width for each athlete. Therefore, these recommendations may help coaches and trainers improve the quality of training and thereby reduce the risk of injury in ski mountaineering.

## Figures and Tables

**Table 1 sports-10-00191-t001:** Demographic and physical characteristic data of the skiers depending on whether they had suffered an injury.

**Characteristics**	**Injured Skiers** **(*n* = 50)**	**Non-Injured Skiers (*n* = 54)**	** *p* **
Age (years)	37.05 ± 13.67	37.33 ± 11.88	0.948
Body mass (kg)	66.81 ± 12.04	68.36 ± 8.98	0.567
Height (m)	1.74 ± 0.07	1.74 ± 0.07	0.990
Body Mass Index	21.94 ± 3.08	22.48 ± 2.28	0.262
Practice ski mountaineering (years)	13.56 ± 11.99	12.64 ± 11.46	0.425
**Characteristics**	**Injured Skiers (*n* = 28)**	**Non-Injured Skiers (*n* = 23)**	** *p* **
Number of competitions per year	1.48 ± 1.7	0.79 ± 1.28	0.046
Number of international competitions per year	1.94 ± 4.85	1.47 ± 4.12	0.400

**Table 2 sports-10-00191-t002:** Characteristics of recorded injuries.

	Total (*n* = 64)	
	*n*%	
Anatomical region		
Foot	9	14
Ankle	7	10.9
Knee	19	29.6
Leg	6	9.3
Thigh	1	1.6
Calf	1	1.6
Pelvis	1	1.6
Iliopsoas	1	1.6
Ribs	1	1.6
Chest	1	1.6
Shoulder	5	7.8
Clavicula	2	3.1
Neck	1	1.6
Face	1	1.6
Finger	1	1.6
Wrist	2	3.1
Hands	4	6.2
Elbow	1	1.6
Side		
Right	34	53.1
Left	27	42.2
Middle	3	4.7
Moment		
Warm up	1	1.6
Training	48	75
Cool down	3	4.7
Competition	12	18.7
Moment of activity		
Uphill	14	21.9
Downhill	42	65.6
Other	8	12.5
Reason		
Impact	44	68.8
Overuse	20	31.2
Diagnosis		
Luxation	5	7.8
Sprain	12	18.8
Tendinitis	4	6.2
Muscle contraction	5	7.8
Fracture	8	12.4
Irritation	2	3.1
Contusion	5	7.8
Fissure	3	4.7
Muscle break	6	9.3
Superficial wound	1	1.6
Other	13	20.4
Time without skiing		
1–3 days	19	29.7
4–7 days	5	7.8
From 1 to 2 weeks	7	10.9
From 2 weeks to 1 month	12	18.8
From 1 month to 3 months	11	17.2
More than 3 months	10	15.6

**Table 3 sports-10-00191-t003:** Training characteristics of skiers depending on whether they had suffered an injury.

Training Aspects	Injured Skiers (*n* = 50)	Non-Injured Skiers (*n* = 54)	*p*
Ski mountaineering training sessions per week	3.10 ± 0.27	2.28 ± 2.19	0.022
Training sessions per day	1.06 ± 0.34	1.09 ± 0.048	0.759
Average session duration (hour)	3.35 ± 0.47	3.03 ± 0.18	0.802
Duration of the longest session (hour)	5.62 ± 0.24	5.94 ± 0.18	0.542
Resistance sessions per week (day)	1.02 ± 0.18	1.17 ± 017	0.359
Months per year of resistance training Core training (days per week) Flexibility training (days per week)	5.57 ± 0.62 1.46 ± 0.20 1.28 ± 0.24	6.56 ± 0.63 2.13 ± 0.25 1.69 ± 0.29	0.223 0.029 0.287

**Table 4 sports-10-00191-t004:** Training characteristics of skiers depending on their number of injuries.

	% Injured Skiers			OR CI 95%	
	Male	Female	OR	LL	UL
Gender	48.7	52.6	0.854	0.312	2.336
	**Yes**	**No**			
Competitor	60	40	0.44	0.195	1.011
Strength training	46.4	53.8	1.346	0.593	3.056
CORE training	45.8	60.9	1.838	0.706	4.788
Flexibility training	47.4	57.9	1.528	0.553	4.220
Warm up	46.7	54.3	1.357	0.588	3.132
Cool down	47.7	53.7	1.253	0.527	2.982

OR = Odds ratio. LL = Lower limit. UL = Upper limit.

**Table 5 sports-10-00191-t005:** Resilience of skiers depending on whether they have suffered an injury.

	Injured Skiers (*n* = 50)	Non-Injured Skiers (*n* = 54)	*p*
Resilience	40.70 ± 0.81	39.27 ± 1.02	0.275

**Table 6 sports-10-00191-t006:** Equipment-related factors depending on whether they had suffered an injury.

Training Aspects	Injured Skiers (*n* = 50)	Non-Injured Skiers (*n* = 54)	*p*
Number of pairs of skis	2.89 ± 1.48	2.23 ± 1.23	0.019
Width of skis (meters)	0.71 ± 12.56	0.76 ± 10.11	0.040
Number of pairs of poles	2.76 ± 1.45	2.29 ± 1.29	0.068
Pole length (meters)	1.34 ± 8.09	1.31 ± 8.30	0.326

## Data Availability

Data are available upon request.

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
