# Peer review of "Relationship of Training Factors and Resilience with Injuries in Ski Mountaineers"

_sports, 2022, doi:10.3390/sports10120191_

Round 1

Reviewer 1 Report

Dear authors, your manuscript is intriguing but I recommend structurally changing the statistical approach because it does not reflect the results it proposes.

Regarding the abstract, I recommend in the methods part to describe the population (years of experience? Age cut off?), The intervention (type of questionnaire, time window of administration) and the outcome (are there other scores or scales used? ). Furthermore, for the drafting of the manuscript I recommend using guidelines:

https://www.researchgate.net/publication/351072130_A_Consensus-Based_Checklist_for_Reporting_of_Survey_Studies_CROSS

Describe the survey in summary. The types of tests used are superfluous ...

Among the keywords "skimo"

In line 50, I suggest adding: "Furthermore, physical exercise consisting of a structured injury prevention neuromuscular warm-up could have an immediate effect in improving the activation time of the knee stabilizer muscles, thus potentially reducing the risk of injury to the knee. "with reference: https://doi.org/10.3390/app11114958

68 but first I would anticipate your concept of resilience as an athlete

80 cross-sectional survey-based study. Experience time in skimo? Amateur, Semi-pro, pro?

82 They are results of inclusion, you need to shift into results. (what part of spain?)

95 is there a score for these characteristics? A scale? Or are they open-ended questions?

97 and [15] the section refers to a textbook in Spanish, is there not a validated or referenced score in the literature on resilience?

Put a copy of the survey as supplemental material

100 when? By what means?

Regarding the statistical analysis, talking about “relationship” concept: have you done a correlation analysis?

120 the response rate? But has an injury occurred in your career?

124-126 I don't understand the relationship .. have you done a logistic regression? YES / NO against the number of training session per week? What test are you conducted?

Again, I do not understand what kind of study is needed to correlate the injury to the number of training sessions, the kruskal does not provide this type of evidence. I recommend a logistic regression ... also because there will be athletes with 5-sessions with injuries or vice versa and you cannot compare the data like this ..

Have 152 suffered an injury in their career?

Author Response

Thank you very much to the reviewers for their contributions to the manuscript with the aim of improving it. Each of the contributions has been responded to below and, for ease of reading, all modifications made to the original manuscript have been marked up using the “Track Changes” function.

REVIEWER 1

Dear authors, your manuscript is intriguing but I recommend structurally changing the statistical approach because it does not reflect the results it proposes.

AUTHORS

Thank you for your input. We have changed our statistical approach.

REVIEWER

Regarding the abstract, I recommend in the methods part to describe the population (years of experience? Age cut off?), The intervention (type of questionnaire, time window of administration) and the outcome (are there other scores or scales used? ). Furthermore, for the drafting of the manuscript I recommend using guidelines:

https://www.researchgate.net/publication/351072130_A_Consensus-Based_Checklist_for_Reporting_of_Survey_Studies_CROSS

Describe the survey in summary. The types of tests used are superfluous ...

AUTHORS

Thank you for your comment. We have modified the information in the abstract, in the methods and materials section and in the first part of the results. We hope we have fulfilled your instructions and those of the article.

REVIEWER

Among the keywords "skimo"

AUTHORS

Thank you for your observation. The word “skimo” has been added to keywords.

REVIEWER

In line 50, I suggest adding: "Furthermore, physical exercise consisting of a structured injury prevention neuromuscular warm-up could have an immediate effect in improving the activation time of the knee stabilizer muscles, thus potentially reducing the risk of injury to the knee. "with reference: https://doi.org/10.3390/app11114958

AUTHORS

Thank you for your suggestion. The sentence has been already added.

REVIEWER

68 but first I would anticipate your concept of resilience as an athlete

AUTHORS

Thank you for your comment. We added the definition of resilience.

REVIEWER

80 cross-sectional survey-based study. Experience time in skimo? Amateur, Semi-pro, pro?

AUTHORS

Thank you for your observation. The skiers who participated in the study had different levels and years of experience, which is why this information has not been detailed. The average number of years of experience is shown in the results section.

REVIEWER

82 They are results of inclusion, you need to shift into results. (what part of spain?)

AUTHORS

Thank you for your comment. In the survey they were asked about their country, not their region, and therefore, we do not have this information.

REVIEWER

95 is there a score for these characteristics? A scale? Or are they open-ended questions?

AUTHORS

Most of the questions were of multiple choice. In the resilience section, a reduced scale was used. Therefore, we have added this information.

REVIEWER

97 and [15] the section refers to a textbook in Spanish, is there not a validated or referenced score in the literature on resilience?

AUTHORS

Thank you for your question. We have found a validation study so we have referenced it.

REVIEWER

Put a copy of the survey as supplemental material

AUTHORS

Thank you for your comment. A copy of the survey will be added as supplemental material.

REVIEWER

100 when? By what means?

AUTHORS

Thank you for your input. We wanted to indicate that in the last section the participants were able to describe all the injuries suffered to date. We have deleted the sentence to avoid misunderstandings.

REVIEWER

Regarding the statistical analysis, talking about “relationship” concept: have you done a correlation analysis?

AUTHORS

Thank you very much for the suggestion. Correlations were calculated between the analyzed different variables, but all of them were close to 0.0 and none were statistically significant. Because it does not provide relevant information to the research conducted, the information has not been included in the final version.

REVIEWER

120 the response rate? But has an injury occurred in your career?

AUTHORS

The response rate was 100%. Among the 104 skiers who completed the questionnaire, some suffered an injury and others did not, so they were divided into two groups according to this criteria.

REVIEWER

124-126 I don't understand the relationship .. have you done a logistic regression? YES / NO against the number of training session per week? What test are you conducted?

Again, I do not understand what kind of study is needed to correlate the injury to the number of training sessions, the kruskal does not provide this type of evidence. I recommend a logistic regression ... also because there will be athletes with 5-sessions with injuries or vice versa and you cannot compare the data like this ..

AUTHORS

Thank you very much for your suggestion. We have combined the two suggestions into one. We have changed our statistical approach in order to avoid misunderstandings. We have performed the Mann-Whitney U test for independent samples to analyze most of the variables according to whether the skiers had suffered an injury or not. The reason we did not perform logistic regression is because we do not assume that causality goes in one direction or the other, but rather contrast the association between variables. Moreover, we are more interested in analyzing separately the relationship of injuries with each of the variables that characterize the workouts. We only used Kruskal-Wallis test to examine the difference between skiers with different numbers of injuries with respect to warm-up and cool-down routines.

Reviewer 2 Report

At the outset, I would like to thank you for the opportunity to review your work.
I consider the article to be very essential from the perspective of competitive and amateur sport.
In this regard, the authors have developed a good diagnosis of the research problem, but only (unfortunately) from the perspective of the trainees.
My comments concern the omission of the specificity of the environment in which skiers practice their sport.
High and medium mountains are associated with a large number of threats.
Skiers have to comply with the accepted mountain safety standards, which (from the environmental perspective) minimize the risk of injury, loss of health and life.
I suggest that you build on your work in introducing and discussing these issues.
I recommend that you refer to work: Piepiora Z, Piepiora P. The snow avalanche event analysis – a proposal of the new method in the example of the Giant Mountains. Arch Budo Sci Martial Art Extreme Sport 2020; 16: 91-104

Author Response

Thank you very much to the reviewers for their contributions to the manuscript with the aim of improving it. Each of the contributions has been responded to below and, for ease of reading, all modifications made to the original manuscript have been marked up using the “Track Changes” function.

REVIEWER 2

At the outset, I would like to thank you for the opportunity to review your work.
I consider the article to be very essential from the perspective of competitive and amateur sport.
In this regard, the authors have developed a good diagnosis of the research problem, but only (unfortunately) from the perspective of the trainees.
My comments concern the omission of the specificity of the environment in which skiers practice their sport.
High and medium mountains are associated with a large number of threats.
Skiers have to comply with the accepted mountain safety standards, which (from the environmental perspective) minimize the risk of injury, loss of health and life.
I suggest that you build on your work in introducing and discussing these issues.
I recommend that you refer to work: Piepiora Z, Piepiora P. The snow avalanche event analysis – a proposal of the new method in the example of the Giant Mountains. Arch Budo Sci Martial Art Extreme Sport 2020; 16: 91-104

AUTHORS

 Thank you for your suggestion. We have added a paragraph (line 49) highlighting the importance of environment-related factors and we have cited the recommended article.

Round 2

Reviewer 1 Report

Unfortunately, I have to recommend in the abstract methods section to describe the population (years of experience? Age cut?), The intervention (type of questionnaire, time window of administration) and outcome (are other scores or scales used?). In addition, for the writing of the manuscript I recommend using guidelines:

https://www.researchgate.net/publication/351072130_A_Consensus-Based_Checklist_for_Reporting_of_Survey_Studies_CROSS

Describe the survey at a glance. The types of tests used are superfluous ...

Unfortunately, the methods section is totally missing .. there is only one goal and immediate results, so it is not adaptable to publication. Moreover, there is an unnecessary description of the tests used which may be omitted

Line 96 - the fact that you have reached 104 people is a result, not a method. The method is where, how and what types of subjects you wanted to reach, all of which is missing. Regarding the population: obese subjects? With comorbidities? Were severe injuries to the upper limbs included?

Line 97 - the fact that most of them come from Spain is a result, not a method. The method is to describe that the questionnaire has been sent to different nations of the European continent.

Statistical Analysis
When I asked to review the statistical approach it was to overturn it (but I recommend it to increase the caliber of the manuscript) .. because if you can't do the correlation then you could do some logistic regressions .. for each variable (continuous) - evaluate injury: YES / NO (nominal)

Author Response

Thank you very much to the reviewers for their contributions to the manuscript with the aim of improving it. Each of the contributions has been responded to below and, for ease of reading, all modifications made to the original manuscript have been marked up using the “Track Changes” function.

REVIEWER 1

Unfortunately, I have to recommend in the abstract methods section to describe the population (years of experience? Age cut?), The intervention (type of questionnaire, time window of administration) and outcome (are other scores or scales used?). In addition, for the writing of the manuscript I recommend using guidelines:

https://www.researchgate.net/publication/351072130_A_Consensus-Based_Checklist_for_Reporting_of_Survey_Studies_CROSS

Describe the survey at a glance. The types of tests used are superfluous ...

Unfortunately, the methods section is totally missing .. there is only one goal and immediate results, so it is not adaptable to publication. Moreover, there is an unnecessary description of the tests used which may be omitted

Line 96 - the fact that you have reached 104 people is a result, not a method. The method is where, how and what types of subjects you wanted to reach, all of which is missing. Regarding the population: obese subjects? With comorbidities? Were severe injuries to the upper limbs included?

Line 97 - the fact that most of them come from Spain is a result, not a method. The method is to describe that the questionnaire has been sent to different nations of the European continent.

AUTHORS

Thank you for your comments. We have included a new sentence about methods in the abstract. Moreover, we have modified the material and methods section, especially the subsections on sample characteristics and data collection methods. We hope we have fulfilled your requests.

REVIEWER 1

Statistical Analysis
When I asked to review the statistical approach it was to overturn it (but I recommend it to increase the caliber of the manuscript) .. because if you can't do the correlation then you could do some logistic regressions .. for each variable (continuous) - evaluate injury: YES / NO (nominal)

AUTHORS

Thank you for your observation. We have added odds ratio analysis and have changed several statements of the article to ensure consistency in the information provided.
